# Assessment Tasks and Virtual Exergames for Remote Monitoring of Parkinson’s Disease: An Integrated Approach Based on Azure Kinect

**DOI:** 10.3390/s22218173

**Published:** 2022-10-25

**Authors:** Gianluca Amprimo, Giulia Masi, Lorenzo Priano, Corrado Azzaro, Federica Galli, Giuseppe Pettiti, Alessandro Mauro, Claudia Ferraris

**Affiliations:** 1Institute of Electronics, Computer and Telecommunication Engineering, National Research Council, Corso Duca degli Abruzzi 24, 10129 Torino, Italy; 2Department of Control and Computer Engineering, Politecnico di Torino, Corso Duca degli Abruzzi 24, 10129 Torino, Italy; 3Department of Neurosciences, University of Turin, Via Cherasco 15, 10100 Torino, Italy; 4Istituto Auxologico Italiano, IRCCS, S. Giuseppe Hospital, Strada Luigi Cadorna 90, 28824 Piancavallo, Italy

**Keywords:** Parkinson’s disease, neurorehabilitation, exergames, azure kinect, UPDRS, movement analysis, body tracking, telemedicine

## Abstract

Motor impairments are among the most relevant, evident, and disabling symptoms of Parkinson’s disease that adversely affect quality of life, resulting in limited autonomy, independence, and safety. Recent studies have demonstrated the benefits of physiotherapy and rehabilitation programs specifically targeted to the needs of Parkinsonian patients in supporting drug treatments and improving motor control and coordination. However, due to the expected increase in patients in the coming years, traditional rehabilitation pathways in healthcare facilities could become unsustainable. Consequently, new strategies are needed, in which technologies play a key role in enabling more frequent, comprehensive, and out-of-hospital follow-up. The paper proposes a vision-based solution using the new Azure Kinect DK sensor to implement an integrated approach for remote assessment, monitoring, and rehabilitation of Parkinsonian patients, exploiting non-invasive 3D tracking of body movements to objectively and automatically characterize both standard evaluative motor tasks and virtual exergames. An experimental test involving 20 parkinsonian subjects and 15 healthy controls was organized. Preliminary results show the system’s ability to quantify specific and statistically significant (*p* < 0.05) features of motor performance, easily monitor changes as the disease progresses over time, and at the same time permit the use of exergames in virtual reality both for training and as a support for motor condition assessment (for example, detecting an average reduction in arm swing asymmetry of about 14% after arm training). The main innovation relies precisely on the integration of evaluative and rehabilitative aspects, which could be used as a closed loop to design new protocols for remote management of patients tailored to their actual conditions.

## 1. Introduction

Parkinson’s disease (PD) is a chronic, disabling neurodegenerative disease characterized by motor dysfunction and symptoms (including tremors, muscle stiffness, bradykinesia, hypomimia, postural changes, gait and balance disorders, and changes in speech and writing) that worsen with time [1].

In recent years, therapeutic advances have lessened the impact of motor disability in daily life, especially in the early stages of the disease [2]. However, motor function is likely to decline as the disease progresses, causing a reduction in autonomy, independence, safety, and perceived quality of life. For example, upper limb impairment makes daily activities more difficult and daunting [3,4], while disorders in balance, posture, and walking increase the risk of falls and injuries [5,6].

In addition to pharmacological therapies, there is growing evidence of the benefits of physical therapy to mitigate the effect of motor symptoms [7,8,9], and recent studies suggest that it should begin early in the disease when motor impairment is still mild [10]. Despite these recommendations, there are still several barriers that make rehabilitation currently underutilized in PD [11,12]. These limitations could be addressed by enhancing monitoring and rehabilitation programs in minimally supervised settings (such as the home), thereby optimizing costs, resources, and access to hospital facilities [13,14].

Recently, much attention has been paid to the potential benefits of exergaming [15,16,17,18,19,20] and virtual reality [21,22,23] for motor rehabilitation and the stimulation of cognitive functions [16,24,25]. The combination of physical and cognitive training for the purpose of stimulating the patient with motor exercises while performing concurrent cognitive tasks, has been also investigated [26,27,28]. In general, exergames are designed to promote movement in virtual game environments, with the aim of exploiting the motivational, rewarding, and fun aspects of videogames, but are adapted to the patient’s current needs and condition, with significant benefits when combined with traditional rehabilitation strategies [29] or cognitive training [30]. However, exergames could be designed for alternative purposes: to become an innovative tool for assessing the patient’s current motor status by stimulating the execution of specific movements to be evaluated, as occurs in clinical practice, while simultaneously rehabilitating/training them in an engaging and fun game environment.

Motor status and severity of impairment in PD are currently and commonly assessed through specific tasks defined in standardized rating scales; the Unified Parkinson’s Disease Rating Scale (UPDRS) is one of the most widely accepted, because the motor examination involves many body regions and functions [31]. Neurologists perform the motor examination during scheduled follow-up visits. However, a more frequent and quantitative assessment of motor tasks would be of great interest for better modulating therapies and design rehabilitation protocols tailored to actual needs and conditions, and to the presence of complications, such as daily fluctuations [32,33,34]. For example, quantitative and automatic assessment of motor tasks in the home environment could overcome the limitations of daily diaries [35,36], allowing objective, frequent, and reliable monitoring of motor conditions and more timely adjustment of treatments. Moreover, it could effectively enable the development of integrated home-based solutions that combine standard motor assessment tasks and rehabilitative exergames, thus improving remote patient follow-up on both aspects [37,38,39].

Over the past decade, several studies have proposed technological solutions to objectively assess motor tasks and analyze specific characteristics of human body motion [40,41]. Wearable devices (including inertial sensors, accelerometers, and smartphones) [42] and optical approaches (including RGB-Depth cameras) [43] have recently been used to implement low-cost and minimally invasive solutions to characterize movement disorders and motor patterns, providing measurements with accuracy comparable to the gold standard systems (such as optoelectronic systems) typically used in clinical or research settings.

Regarding PD, wearable sensors have been used to measure gait parameters and freezing [44,45], assess motor symptoms [46,47,48,49] and fluctuations [34], and characterize specific tasks, such as sit-to-stand and stand-to-sit [50]. In general, solutions based on wearable sensors are less practical and technically more complex to manage, especially in terms of calibration and pairing procedures. In addition, sensors placed on body segments could be uncomfortable and intrusive for natural movements, especially in the case of short segments, such as fingers [51].

Optical approaches leverage video-recording devices (e.g., cameras) and vision techniques to implement completely non-invasive and easily manageable solutions for tracking and analyzing human body movement. Several studies have proposed optical approaches to characterize upper limb motor function [52,53], analyze lower limb dysfunctions and postural control [54], estimate gait features [55,56], evaluate arm swing [57,58], and analyze balance disorders [59,60]. Optical approaches have also been widely used in motor rehabilitation for specific pathologies [61,62], including PD [15,63], due to their portability, versatility, high usability, and easy integration into virtual environments. Examples of rehabilitation using optical approaches include upper and lower limbs [64,65], balance disorders [66], cognitive and motor dysfunction [67], and walking [63]. However, to our knowledge, there are no technological solutions for PD that uses optical approaches to simultaneously address quantitative assessment of motor status and its rehabilitation/training, especially suitable for home settings.

With this goal in mind, we designed a solution based on the Microsoft Azure Kinect DK sensor that meets both assessment and rehabilitation/training purposes through a subset of evaluative motor tasks and exergames in a virtual environment. Many recent works have already investigated this new sensor through validation protocols against gold reference systems [68,69,70], agreeing on its higher accuracy and performance compared to predecessors and other commercial optical sensors. Other studies, instead, focused on its new body-tracking algorithm to verify its accuracy, robustness, and reliability in capturing 3D movements and poses [71,72,73]. The Azure Kinect has also recently been used in some preliminary clinical studies and for rehabilitation purposes [74,75,76].

Based on these findings, in the proposed solution we have used the new Azure Kinect to quantify motor features related to postural control, lower limb movement, and walking, as in [54,58]. In addition, we have designed and integrated some rehabilitative/training exergames to stimulate motor control and coordination by eliciting repetitive limb movements that are usually impaired in PD [77]. Specifically, the system includes three tasks derived from the UPDRS (leg agility, gait, and postural stability) and three exergames in a virtual game environment. The latter solicit movements of the upper and lower limbs to improve limb mobility. The exergames are configurable by tuning some game parameters, and consequently changing the difficulty of the exercises, stimulating each patient appropriately according to the current motor status and rehabilitation goals defined by the therapist.

An experimental campaign was organized involving healthy volunteers and subjects with PD who used the proposed solution under the same conditions and constraints. The study had the following objectives: to verify the system’s ability to measure features related to movement, detecting differences between healthy and parkinsonian subjects; to verify whether exergaming could be further combined with motor evaluation, as a more stimulating alternative to standard motor tasks; and to check whether the system is able to detect immediate changes in upper limb mobility during walking after performing exergames for arm extension training. Preliminary results on these goals are presented and discussed.

## 2. Materials and Methods

### 2.1. System Design and Human–Computer Interaction

The proposed system includes a few hardware components (mini-computer, Azure Kinect sensor, and monitor) that implement a simple, non-bulky, and contactless solution for 3D motion capture, and thus are also suitable for home environments. The mini-computer is a ZOTAC© (Zotac, Fo Tan, New Territories, Hong Kong, China) ZBOX EN52060-V model equipped with a 9th generation Intel^®^ Core^TM^ processor (2.4 GHz quad-core), 16 GB of RAM, NVIDIA GeForce RTX 2060 6GB GDDR6, HDMI and USB3 ports, Windows 10 operating system. The high-performance configuration is necessary to meet the hardware requirements of the body-tracking algorithm [78], thus enabling real-time tracking and processing of human body movements. Among the available camera operating modes [68], we selected 1080p resolution for color streaming, Narrow Field of View (NFW) for depth streaming, and 30 frames per second for both. The overall setting of the instrumentation is the same as in [79].

The low-layer software is based on the Azure Kinect Sensor and Body Tracking SDKs, version 1.4.1 and 1.0.1 respectively [80]. The first SDK includes the functions of interfacing with the device and acquiring video streams, while the second includes the functionality of body tracking. The body tracking algorithm integrates Deep Learning (DL) and Part Affinity Field (PAF) [81] approaches to estimate 32 joints of a 2D skeletal model, which is then augmented to 3D in real time using depth information and predictive algorithms trained on real and synthetic images [78].

Human–computer interaction (HCI) and graphical user interface (GUI) are key elements to support the management of a system and ensure high usability, especially in the case of elderly with disabilities [82]. To this end, the same principles for HCI and the same GUI design described in [79] were also employed for this study, using the 3D skeletal model to characterize the motor function and also to interact with the system naturally and intuitively through body movement. The GUI of the evaluative motor tasks and the three exergames were implemented in Unity^®^ (Unity Technologies, San Francisco, CA, USA). All activities were performed in front of the 3D camera to ensure the accuracy of the depth sensor and skeletal model prediction [69,83].

### 2.2. Assessment of the Motor Condition: The Evaluative Motor Tasks

In general, the evaluative motor tasks aim to characterize the patient’s motor condition through quantitative functional parameters. The subset implemented in our system is suitable for safely assessing lower limb mobility, gait, and postural stability even in home and minimally supervised settings. The following tasks have been considered:Leg Agility (LA): UPDRS task to assess the impairment of motor control and coordination in the lower limbs, typically affected by PD symptoms, through repetitive leg movements performed separately with the left and right leg.Postural Stability (PoS): a 30-s balance task to assess stability in the standing position through the swaying of the body’s center of mass (COM) estimated from the skeletal model, as in [54]. This task is indeed less risky than the traditional UPDRS pull-test, especially in home settings. However, the strong correlation between COM sways and postural instability and gait difficulty (PIGD) score has been previously verified [54].Gait (G): UPDRS task to assess gait disorders through some spatiotemporal and arm swing parameters on a short walking path, as in [56,58,84].

### 2.3. Rehabilitation/Training of the Motor Condition: The Virtual Exergames

Exergames usually aim to stimulate specific motor functions in a more engaging, motivating, and enjoyable way than traditional physical training. The proposed system offers three exergames to stimulate upper and lower limb movements through extension and mobility exercises. The following exergames were developed:Lateral Weightlifting (LWL): this exergame is performed in a standing position and consists of a sequence of lateral arm lifts. The exercise aims to strain the flexibility, agility, and mobility of the upper limbs. The exergame is set in a gymnasium scenario and mimics weightlifting to engage patients in pseudo-real physical activities.Frontal Weightlifting (FWL): this exergame is similar to the LWL since it stimulates motor control, coordination, and muscle tone by promoting trunk and arm extensions in the same scenario. In contrast to LWL, the exercise consists of a sequence of frontal arm lifts.Bouncing ball (BB): this exergame relies on repetitive movements of the lower limbs to stimulate motor control and coordination through leg mobility. The exercise aims to stress lower limb agility, thus counteracting balance dysfunctions and gait disorders. The exergame is set in an office scenario and consists of dribbling a ball with the legs (thighs), mimicking the movements of the LA task. The exercise is performed in a sitting position only.

A configuration file is available for each exergame to set specific game parameters, thus changing the exercise difficulty. For each game parameter, three possible values were defined (EASY, MEDIUM, HARD) to allow the clinician to set the difficulty of the exercise according to the patient’s motor condition and rehabilitation goals. The complete list of game parameters is shown in Appendix A.

Figure 1a shows the main GUI of the LWL and FWL exergames. The exercise consists of repeating a predefined number of lateral lifts (or frontal for FWL) first with the right arm, then with the left, and finally with both arms simultaneously, according to the ARM_MOV_ parameter. The arm movement, collected through the skeletal model, is mapped on the arm displayed in the game scene. The patient is asked to perform the planned lifts within a maximum time (ARM_MAXTIME_) to complete the game. Each lift is analyzed in real time to count the good movements, in which the arm angle exceeds the minimum required threshold (ARM_MINANG_). Good movements are associated with positive acoustic feedback and awarded with a game point. On the contrary, incomplete or partial movements (poor movements, PM), movements in the wrong direction (e.g., frontal instead of lateral and vice versa), or asynchronous movements (in the level of simultaneous lifts), are associated with negative acoustic feedback and assignment of an error.

Figure 1b shows the main GUI of the BB exergame. The exergame consists of repeating a predefined number of lifts of the right leg and then of the left leg to hit a ball, according to the LEG_MOV_ parameter. The ball is highlighted by a halo when ready to be hit, and the patient should wait for the halo “light-on” before starting the movement. The starting position of the ball is set as a percentage above the knee rest position (BALL_START_): when hit, the ball bounces upward, then it falls back to the starting position. The time between a ball hit and the next ball light-on is controlled by the LIGHTON_TIME_ configuration parameter, which therefore determines the cadence of the game execution. The leg movement, collected from the 3D skeletal model, is mapped on the avatar’s corresponding leg. The patient is asked to perform the planned lifts within a maximum time (LEG_MAXTIME_) to complete the level. Each lift is analyzed in real time to count the good movements, i.e., those that lead to hitting the ball. Good movements are associated with positive acoustic feedback and awarded with a game point. PM or movements starting before ball light-on are associated with negative acoustic feedback and the assignment of an error.

### 2.4. Participants and Experimental Protocol

Because the system is specifically designed for home use by PD patients who require limited supervision, we expect that patients with an Hohen & Yahr (HY) score ≤3, i.e., mild to moderate disability, may benefit from the proposed solution, as monitoring of changes in motor condition and engagement in rehabilitation are critical at these stages. In addition, the severity of motor impairment does not yet preclude the safe use of the system with only, for example, caregiver supervision. Therefore, an experimental protocol was established with the following exclusion criteria: severe disability (HY > 3), severe and near-permanent tremor with inadequate response to treatment, cognitive impairment (Mini–Mental State Examination Score <27/30), and severe visual impairment.

The experimental study was organized in a supervised setting to obtain preliminary feedback on the system before using it in home settings. A group of 20 subjects with PD was enrolled at the Division of Neurology and Neurorehabilitation of San Giuseppe Hospital (Istituto Auxologico Italiano, Piancavallo, Verbania, Italy). A group of 15 healthy subjects, without neurological and cognitive disorders, was also recruited as a control group (HC). All participants signed a written informed consent before participating in the study, which was approved in advance by the local ethics committee according to the Declaration of Helsinki (1964) and its amendments.

The experimental protocol, schematically described in Figure 2, consisted of two trials for each participant, with a 15-min break between them. In the first trial, participants were instructed on the use of the system and tried some basic interaction with it, to become familiar with motor tasks and exergames. The second trial was used to collect data and estimate functional parameters. During this trial, participants were requested to perform the sequence of evaluative motor tasks (LA, PoS, and G) before (PRE session) and after (POST session) the three exergames (LWL, FWL, and BB). The POST session was included to investigate the ability of the system to detect changes in motor performance during the evaluative motor tasks due to the previous execution of the exergames. This was expected especially for LWL and FWL with respect to G, as the two exergames were designed to improve arm mobility, and therefore could have an immediate effect on arm swing during walking even after a single trial (see Section 3.5). In addition, the POST session was used to obtain additional data for the comparison between the traditional LA and its gamified dual, the BB exergame (see Section 3.4). All participants were involved for about 1 h, including breaks between trials and single tasks, and performed the activities in the same order and under the same condition, as indicated by the experimental protocol.

### 2.5. Objective Characterization of the Motor Performance

The analysis procedure automatically estimates functional parameters from the 3D skeletal model using joint trajectories and angles between pairs of body segments. For LA, the estimated parameters refer to leg angles (ANG_LEG_). For PoS, the sway of the body center of mass (COM_BODY_) is estimated from the skeletal model, as in [52,85,86], as the weighted average of the 3D centroids of some body segments, considering their mass and length as indicated by the anthropometric tables related to the anatomy of the human body [87]. For G, the estimated parameters are a subset of traditional spatio-temporal measures and arm swing parameters [56,58]; arm swing is a crucial element of walking, and its impairment is often evident in individuals with PD.

The objective characterization of the LWL and FWL exergames mainly concerns the elbow (ANG_ELBOW_) and arm (ANG_ARM_) angles. ANG_ARM_ is the angle between the shoulder/wrist and the trunk segments, while ANG_ELBOW_ is the angle between the shoulder/elbow and elbow/wrist segments. The exercise starts with the arms extended along the body in a standing position. Then, depending on the game subtask, one or both arms must be lifted as high as possible, performing repetitive lateral or frontal abduction/adduction movements of the arm, while keeping the elbow in extension. Therefore, the exercise implies continuous and relevant variations in ANG_ARM_, but no variations in ANG_ELBOW_: any deviation from this behavior denotes unexpected and incorrect execution of the exercise. The analysis procedure segments the ANG_ARM_ trajectory into minimum–maximum-minimum cycles, corresponding to each arm movement, to estimate specific kinematic parameters related to angular peaks, duration of cycles, and movement velocity. In addition, the analysis estimates the mean value of ANG_ELBOW_, to highlight difficulties in keeping the arm in extension during the exercise, and an index of simultaneity (INDEX_SIM_) to highlight difficulties in performing simultaneous and coordinated movements. About INDEX_SIM_ of LWL and FWL, this metric was defined based on the collected data to evaluate the time lag between left and right ANG_ARM_ signals at arm raise above ARM_MINAG_ and descent below ARM_MINANG_. Its value is close to 0 for good time alignment in the relevant motion instants; otherwise, it becomes larger according to the desynchronization level. Single arm raises, if present, penalize the result through amplification of the value. A more rigorous description and formula to compute the INDEX_SIM_ parameter is reported in the Appendix A. The parameters for LWL and FWL were estimated for each subtask of the single execution separately (left arm lifts, right arm lifts, and simultaneous left and right lifts).

The BB exergame stimulates lower limb movements, so the objective characterization concerns leg (ANG_LEG_) and knee (ANG_KNEE_) angles. ANG_LEG_ is the angle between the hip/knee and trunk segments, while ANG_KNEE_ is the angle between the hip/knee and knee/ankle segments. The exercise starts with the arms resting sideways in a sitting position. Next, the leg must be raised to hit the ball, performing repetitive up and down movements of the leg with the knee flexed. Thus, the exercise involves continuous and relevant variations in ANG_LEG_ during the execution of movements, but no or minimal variations in ANG_KNEE_; any deviation from this behavior denotes unexpected and incorrect execution of the exercise. The analysis procedure segments the ANG_LEG_ into minimum–maximum-minimum cycles, corresponding to each up/down leg movement, to estimate specific kinematic parameters related to maximum angular peaks, leg excursions, cycle duration, movement velocity, and frequency. In addition, the analysis estimates the mean value of ANG_KNEE_ in order to highlight difficulties in controlling knee flexion during the exercise. The parameters for BB were estimated for each subtask of the single execution separately (left leg lifts and right leg lifts). Since the BB exergame aims to mimic the LA evaluative task, the same functional parameters were considered to effectively compare the two exercises, which is one of the goals of the study.

The complete list of functional parameters computed for evaluative motor tasks and virtual exergames is shown in Table 1.

Although for game scoring, the system analyzes angular trajectories in real time, the characterization of motor performance through functional parameters was performed through an in-depth analysis procedure. This analysis procedure consisted of custom-written MATLAB^®^ (Mathworks Inc, Natick, MA, USA) scripts that automatically extracted functional parameters from the collected data stored as JSON files. Data pre-processing was based on resampling and filtering procedures. Three-dimensional joint trajectories were resampled at 30 Hz to remove typical framerate jittering. Then, a third-order Butterworth low-pass filter was applied to the resampled data to remove high-frequency noise. Only the spectral band below 8 Hz, relevant for human body movements, was retained without considering the high-frequency components that were not significant for this study. Next, the analysis procedure determined the functional parameters from the preprocessed data for each exercise. To provide an intuitive and easy-to-interpret comparison of motor performance, a graphical representation of the functional parameters using radar charts was employed. Since the functional parameters represent quantities with different scaling and magnitude, a min–max normalization was applied, considering the maximum and minimum values of parameters among all the participants’ performances to enable their simultaneous visualization (see Section 3).

### 2.6. Statistical Analysis

The functional parameters used to characterize the evaluative motor tasks and the exergames were statistically analyzed separately for PD and HC to identify similar or different trends in the two groups.

To determine whether to proceed with a parametric or nonparametric statistical analysis approach, the normality distribution was evaluated for each of the estimated parameters. For this purpose, given the small number of sessions collected, we used the Shapiro–Wilk test. The results of the test showed a deviation from the normality condition (*p* < 0.05) for all parameters in Table 1. Therefore, a nonparametric approach was selected for statistical analysis, and the median and percentiles were computed and compared.

Next, the distributions of the extracted kinematic features for the two groups were objectively compared through the nonparametric Mann–Whitney U test for independent samples to detect statistical differences between their distributions.

Statistical analysis was performed using Jamovi (version 2.2.5), an open-source modular platform for statistical computing [88]. A 95% significance level (*p* < 0.05) was considered for statistical tests, which is a widely accepted criterion for clinically meaningful evidence in research against the null hypothesis.

## 3. Results

### 3.1. Group Characteristics and Data Collection

The participants of the PD group had the following characteristics: 71.1 ± 9.2 years (average age), 8.1 ± 6.8 years (average disease years), 33.7 ± 5.9 pts (average UPDRS score), 2.2 ± 0.9 pts (average H&Y score), and nine females and eleven males (gender). The healthy controls in the HC group had the following demographic characteristics: 68.8 ± 5.9 years (average age), and seven females and eight males (gender).

All participants completed the planned sessions for data analysis. Nevertheless, not all PD participants were able to successfully complete all the exercises in each session (PRE and POST) due to an impairment that hindered specific movements. In this case, the technical operator stopped the exercise to avoid stress or excessive strain on the patient. Hence, these executions were discarded, as well as those invalid due to external interference during the acquisition phase (e.g., presence of other people, subjects talking while performing the exercises, subjects becoming distracted).

Table 2 summarizes the number of completed and valid executions considered for the data analysis. The reported values for evaluative tasks aggregate PRE and POST executions. For LWL, FWL, LA, and BB, the subtasks involving only one side of the body (i.e., left side and right side) or both sides simultaneously (only for LWL and FWL) were considered as independent executions.

### 3.2. Statistical Analysis of the Evaluative Motor Tasks

The results of the statistical analysis for evaluative motor tasks are shown in Table 3, which reports median, 1° and 3° percentile values of the functional parameters, estimated separately for PD and HC groups on all the collected valid executions. In addition, Table 3 also reports the results of the Mann–Whitney U test along with the *p*-value and significance level. This analysis aimed to highlight the ability of the system to objectively characterize the motor performance of the subjects in performing the evaluative tasks and to identify significant differences and similarities in functional parameters among the two groups involved in the study.

The comparison between PD and HC groups highlights some differences, as expected. Concerning LA, the median values of parameters related to spatial properties of the movement (mean excursion angle, sway areas, and volumes of KNEE and ANKLE joints), and parameters related to velocity and frequency are higher in the HC group, with a wider range also between 1° and 3° percentiles. As expected, this is explained by the better mobility of healthy participants on average, with respect to general muscle stiffness and reduced mobility typical of parkinsonians. These findings were confirmed by the *p*-values of the Mann–Whitney U test, which identified all the features as discriminating between the two groups but for EXC_M_ and KNEE_M_. The former was close to the significance value (*p* = 0.062), suggesting that additional data including more pathological PD subjects could move this parameter below the threshold value as well. On the other hand, KNEE_M_ was not significant, probably because most of the recruited PD subjects did not show dyskinesias or severe impairment in controlling knee flexion in this body position.

Concerning postural stability, the PD group showed more instability during the 30-s standing task: this was highlighted by a greater range of swaying of COM_BODY_ in AP and ML directions, covering a more relevant sway area. This was confirmed by the statistical test, for which the null hypothesis of equal distributions was rejected for all parameters but ML_T_ and ML_S_. Indeed, the median values of these two parameters were closer in the two groups. This result was probably due to the type of standing position with parallel but slightly spaced feet: this likely promotes a greater imbalance in the AP direction rather than in the ML direction. In addition, no patients with severe balance problems were involved, which would likely have enhanced the results for ML parameters. However, even if the differences in the values of the PD parameters were globally small compared to HC, because the recruited PD participants had only mild balance disorders, it was important to note that fine differences had been detected by the system.

Finally, the spatio-temporal parameters related to the G task confirm the findings identified in the previous tasks. The PD group was characterized, as expected, by longer double support (DSUP) and stance in the gait cycle (STANCE), and reduced step length (STEP_L_) and walking speed (SPEED) compared to HC. Only cadence was similar between the two groups (*p* = 0.144), probably due to the short length of the walking path, which did not allow for the highlighting of significant differences for the CADENCE parameter. The median values of arm swing parameters also distinguished the two groups. The HC subjects were characterized by higher arm speed and sway ranges along the three directions and less asymmetry than PD subjects, which was explained by the overall better mobility of the upper limbs of these subjects. The statistical test supported these results, except for those of ML_ARM_R_. However, this direction was not the most distinctive of arm swing motions, which is mainly characterized by a movement along the AP and UD directions, so we did not expect significant differences along ML.

As previously mentioned, the radar charts provide an immediate and intuitive comparison between healthy and pathological performance. Since the functional parameters correspond to different physical quantities, a minimum–maximum normalization on the range [0, 1] was applied to provide a clearer and more effective graphical representation. To this end, for each parameter, we considered the best parameter value (that could be the maximum or minimum value on all the collected trials according to the parameter meaning) as associated with the maximum normalized value (i.e., 1) and the worst parameter value among the collected trials was associated with the minimum normalized value (i.e., 0). This procedure was repeated for each parameter and every evaluative motor task. It is important to note that some functional parameters directly link with the severity of the impairment (i.e., the parameter values increase with increasing motor dysfunctions). According to the convention used for the radar graphical representation, the worsening of these parameters (i.e., increasing parameter values) correspond to normalized values closer to zero. On the contrary, other parameters show an inverse relationship with the impairment severity, so their values decrease with increasing motor dysfunctions. In this case, worsening parameters (i.e., decreasing parameter values) correspond to normalized values closer to zero in the graphical representation. In this way, all the normalized parameters are inside the [0, 1] range, so generating radar charts that expand outward when the motor performance is “good” and shrink inward when the motor performance is “bad”.

Using this convention, Figure 3 shows the difference between the average parameters estimated for the two groups of subjects on the three evaluative motor tasks.

### 3.3. Qualitative and Statistical Analysis of Virtual Exergames

Similarly, the proposed solution uses training sessions to evaluate the motor performance of the upper and lower limbs during the virtual motor exergames provided by the system.

Some 3D trajectories collected during the virtual exergames are here presented and discussed, thus demonstrating the potentialities of the proposed solution.

The first example refers to the first subtask of the LWL exergame, which involved five lateral abduction–adduction movements (ARM_MOV_ = 5) of the left arm to complete the level. Figure 4 shows an example of the ANG_ARM_ trajectory. The analysis procedure classified all movements as good movements because the maximum angle peaks were all above the pre-established minimum threshold. In addition, the number of total movements performed was sufficient to complete the game level. This positive outcome suggests that this subject could be challenged with an advanced parameter configuration in a hypothetical new session e.g., increasing the value of threshold ARM_MINANG_ and/or the number of required movements (ARM_MOV_).

The second example refers to the simultaneous subtask the FWL exergame, which involved five frontal abduction–adduction movements (ARM_MOV_ = 5) of both upper limbs. Figure 5 shows the ANG_ARM_ trajectories of both arms. In this case, the analysis procedure detected anomalies in the motor performance: one poor movement (PM), i.e., a peak of the angle lower than the pre-established threshold, and two non-simultaneous movements in which only one arm was raised.

The third example (Figure 6) refers to the ANG_LEG_ trajectory, collected during the BB exergame. The exercise involved five up-down movements (LEG_MOV_ = 5) of the right leg to complete the subtask. The graph also highlights the instants in time in which the ball was ready to be hit (light-ons) and the ones in which a ball/knee collision verified (hits). As can be appreciated, after the first hit, the subject started repeating the movement before the ball light-on, generating three errors. However, he eventually learned to coordinate his movements to hit the ball only as soon as the halo appeared again, as in the last movement.

From these results, it is reasonable to conclude that the functional parameters, estimated from angular trajectories, may objectively characterize the motor performance of the training sessions and detect alterations, as occurred for evaluative motor tasks.

In addition to the qualitative analysis provided by the graphical representation, the results of the statistical analysis of the extracted functional parameters are reported in Table 4. The comparison between PD and HC groups highlights some relevant differences.

Concerning BB, median values of parameters related to spatial properties of the movement (mean excursion angle, sway areas, and volumes of KNEE and ANKLE joints) and velocity/frequency parameters were higher in the HC group, with a larger range also between the 1° and 3° percentiles, as already observed in LA. Only the maximum frequency (F_MAX_) was similar among the two groups, but this was coherent, considering that the pace of the game was fixed by the LIGHTON_TIME_ parameter of the game for both the PD and HC groups. Nevertheless, the B90 parameter highlighted how the PD group was characterized by a distribution of spectral power mainly in lower frequencies with respect to HC. These findings were confirmed by the *p*-values of the Mann–Whitney U test, which identified all the features as discriminating but for F_MAX_ and KNEE_M_. For the latter, the same explanation proposed for LA analysis holds.

Concerning LWL and FWL, as expected, angle excursions and movement speed were reduced in PD group, which was also characterized by a longer duration (DUR_M_) of the single arm raises, as well as the overall time to complete the whole task (EXTIME). The defined synchronicity index (INDEX_SIM_) for evaluating simultaneous lifts proved as well to be a discriminating factor between the groups, with a reduced value in HC (i.e., greater temporal alignment between the arms). These findings were supported again by the statistical test results. Only ELBOW_M_ did not reject the null hypothesis (*p* > 0.05), but the recruited PD subjects did not exhibit specific impairment in maintaining the upper limb in an extended position during motion, which was reflected by very similar median values between the two groups.

As for evaluative motor tasks, Figure 7 reports a graphical comparison, as normalized radar charts, between the average values of the functional parameters of the exergames in PD and HC groups.

### 3.4. Exergaming as Alternative or Complementary Evaluation

As described in Section 1, exergames may represent an innovative and alternative strategy for rehabilitation, that exploits the positive psychological stimulus of gamification to increase user engagement and satisfaction, which are fundamental aspects to ensure continuity of use. This feature is crucial for telemedicine, as patients could soon lose interest in the solution if not properly engaged. Therefore, it could be of interest to also gamify traditional motor assessment to solicit longitudinal usage by subjects and consequently enable continuous monitoring over time. Indeed, rehabilitation exergames themselves may be seen as an additional source of information about the patient’s current motor status, as the movements performed to complete the games may be collected and further analyzed, as happens in the proposed system.

In particular, the BB game was designed as a companion to the LA assessment task, since the same actions (i.e., repeated upward and downward movements of the bent leg) are required to successfully complete the game. Moreover, if the game configuration parameters are properly set, BB could solicit an execution of the leg movements similar in their kinematic signature to those of LA (i.e., the same rhythm and spatial excursion).

To investigate whether such property was achieved by gamification, we compared the mean values of the functional parameters in LA and BB through the same radar chart representation proposed in the previous subsections. We restricted this analysis to the PD group only, as this is the real target of our solution. The result is reported in Figure 8.

As can be appreciated, the two graphs are almost perfectly overlapping in terms of spatial parameters, especially those related to sway areas and volumes of ANKLE and KNEE joints. On the other hand, the parameters related to velocity and cadence (frequency) of movement execution (F_MAX_, SPEED_M_, B90) were quite different. This could be explained by several factors. First, since a small number of possible values (three) were defined for the LIGHTON_TIME_ parameter, it is likely that these predefined levels were not adequate to stimulate, in all subjects, responses comparable to the ones they would have during traditional LA. Secondly, it should be considered that the restricted gamified setting (i.e., fixed ball initial position and time between consecutive ball light-ons) limited the natural variability typical of the LA task, causing a more constrained and similar series of leg lifts. Moreover, in the traditional LA, the subject receives no visual or acoustic feedback about his performance, whereas during BB, he has immediate feedback from the motion of the virtual avatar and the assignment of a positive score. Indeed, as we observed, different subjects tended to have a different response to game. With an almost even distribution, some subjects were completely unaffected by the gamified setting of BB, performing an execution very similar to LA when the game parameters were appropriate for the patients’ motor conditions; on the other hand, other subjects, even when the game parameters were appropriate, tended to alter their normal execution; for instance, raising their leg much more than required to hit the ball, as they “feared” that their movement would not be sufficient to complete the task (i.e., EXC_M_ is indeed larger on average in BB than in LA). Examples of these two observed behaviors are reported in Appendix A, included in the Appendix A.

### 3.5. Exergaming as Mobility Training and Rehabilitation

Even though the protocol employed for this preliminary evaluation of the system did not allow for a longitudinal assessment of the effect of rehabilitation through the proposed exergames, we wanted to test whether the immediate effects of LWL and FWL could be detected by the system in the upper limb mobility during the G task. Indeed, LWL and FWL were designed to promote arm mobility, hence we expected a benefit for the arm swing during walking. For this purpose, we compared the mean values of arm swing parameters (G task) measured before executing the two exergames (PRE session) and their values after training (POST session). Again, we restricted this analysis to the PD group only, as this was the real target population of our solution.

Figure 9 reports, in normalized values for simultaneous visualization, the variation of arm swing parameters in the two sessions. As highlighted, even after a single execution of LWL and FWL exergames, the effects were quite evident. The asymmetry index (ASA_AP_S_) showed a significant reduction in the POST session (more than 10%) that meant more symmetry in the arm swing during walking after LWL and FWL exergaming. Moreover, the arm swing range in the three directions (AP_ARM_R_, ML_ARM_R_, UD_ARM_R_, MAX_ARM_AR_) and arm speed (SPEED_ARM_) increased, suggesting an overall improvement in upper limb mobility during walking in the POST session.

## 4. Discussion

The frequent assessment of the motor condition in PD subjects, daily and in unsupervised settings, could assume an important role for clinicians to tailor and optimize therapies and treatments according to the actual patients’ needs. This aspect has become more critical as growing evidence supports the positive role of early and continuous rehabilitation in PD [10].

In PD, traditional rehabilitation methods focus on improving and optimizing residual motor functions, acquiring alternative strategies, and preventing severe consequences (such as falls and injuries). However, these approaches require several rehabilitation sessions to be effective in terms of number and frequency. In this scenario, the continuous and frequent assessment of motor conditions is still a problem for PD management, both for pharmacological and rehabilitative purposes. In fact, it requires many scheduled specialist visits that unfortunately could soon become unsustainable in terms of costs and resources. Therefore, new services able to provide a quantitative and comprehensive overview of PD patients’ clinical conditions, especially in home settings, could help to modulate and target the proper treatments, as required. These solutions could also lead to personalized rehabilitation protocols, according to patients’ actual needs, and suitable for unsupervised settings, thus ensuring continuity of rehabilitative exercises and reducing patients’ discomfort, long-term complications, and the lower efficacy of non-optimized treatments.

The solution that we propose aims to take a step forward by integrating the objective and automated assessment of the motor condition, conducted through evaluative motor tasks, with specific rehabilitation/training activities comprising motor exercises in engaging virtual environments (exergames). From the perspective of this integration, the proposed exergames not only allow for daily and self-managed training but can be used to collect additional data to assess health status and even provide immediate effects on the motor condition, which may be objectively measured by the evaluative motor tasks themselves. This could be the first step toward defining new future treatment strategies based on a closed loop between evaluative motor tasks and rehabilitative exergames. This innovative approach would allow, on the one hand, rehabilitative exercises to be automatically adapted to the current motor condition measured through the evaluative motor tasks and, on the other hand, to quantify the effects of the rehabilitative exercises on the motor condition, thus allowing the rehabilitation protocol to be tailored to achieve new rehabilitation goals.

To pursue this goal, the proposed solution adopts a user-friendly and non-invasive approach to capture body movements in real time, suitable for minimally supervised settings and based on a single RGB-Depth sensor (Azure Kinect).

For the evaluation of the motor condition, the three tasks (LA, PoS, G) derived from UPDRS were chosen because they were clinically relevant for assessing the impairment of specific motor functions (motor control, stability, agility, coordination, synchronization) and were suitable for being performed safely at home, unattended, or with minimal caregiver supervision. Thus, the proposed solution provides three exergames in a virtual environment that stimulate the motor function of the upper (LWL and FWL) and the lower (BB) limbs for rehabilitation purposes, to train and improve some common dysfunctions occurring in PD patients. It is important to note that the exergames also intrinsically include cognitive aspects, such as the performance of strictly frontal or lateral movements with the arms, or hitting the ball at the exact light-on moments, which provide additional challenges during rehabilitation. In addition, the exergames are configurable (Appendix A) to meet, on one side, the needs of different impairment conditions and, on the other, to stimulate patients to reach new rehabilitation goals.

The preliminary results indicate that the functional parameters defined both for the evaluative motor tasks and for the exergames (Table 1) succeeded in characterizing motor performance for evaluative and rehabilitative purposes through quantitative measures, thus capturing movement features and anomalies (Figure 3, Figure 4 and Figure 5). In addition, the functional parameters suggest that the system can detect even small, but significant, differences between PD and HC subjects.

Regarding the further potential of exergames described in Section 3.4 and Section 3.5, the analyses show the advantages of merging together evaluation and rehabilitation tasks in an integrated system. For example, even if not able to completely replace the traditional LA, the BB game suggests that the gamification of the evaluation stage could elicit the activation of motor strategies that are not put in place by subjects in traditional assessment. For instance, subjects who prioritize speed over leg excursion in LA could be solicited by a higher ball starting position in BB to assess the subject’s real excursion ability, or vice versa, by soliciting a faster execution in BB but with less demanding amplitude. In this perspective, an initial evaluation of the traditional LA task by the system could be a starting point to automatically set the configuration parameters for a subsequent BB execution, facilitating rehabilitation and training suited to the current motor condition. We plan to further explore the automatic, evaluation-based configuration of exergames in future works.

Moreover, even though we have not yet evaluated the long-term benefits of rehabilitation/training through the proposed exergames, we have shown that LWL and FWL can provide immediate benefits to the mobility of the upper limbs during G. This effect was objectively identified by asking subjects to repeat G after the two exergames. This aspect could be exploited, for example, to automatically verify whether the rehabilitation session was successful, and whether the rehabilitation goals need to be modified by the clinician (or automatically by the system itself) in subsequent rehabilitation sessions. This aspect will be further investigated as well in future works.

In addition to these results, the significant number of valid executions collected for data analysis, and the fact that all subjects, after minimal training, were able to complete the whole trial with the system, suggest the ease of use of the system by both PD and HC subjects and its suitability for home environments.

From the perspective of deploying the proposed solution in home settings, the automated detection of changes in habitual motor behavior due to a more significant impact of symptoms on motor performance becomes possible by configuring the assessment sessions (which include the evaluative motor tasks) and the rehabilitation sessions (which include the exergames) at pre-scheduled times of the day or when necessary. This facility could therefore address the issues related to unpredictable motor fluctuations during the day and the uncertainty of self-reported diaries, and could ensure the continuity of rehabilitation programs.

The objective and quick comparison of motor performance becomes, in this way, feasible through quantitative measures and graphs that could directly support clinicians in easily and remotely monitoring of patient’s condition. We expect that the data collected in this scenario will confirm the preliminary results presented in this study.

Furthermore, it is important to note that the design of the proposed system is modular and scalable. The possibility of including additional assessment tasks and virtual exercises, which stimulate new movements and motor functions, makes the experience increasingly satisfying for patients and more comprehensive for clinicians. In addition, this strict synergy could also represent the optimal approach to combining cognitive and physical training, by soliciting patients to perform movements while executing an exergame at different cognitive loads. This aspect is essential in advanced patients with cognitive impairment [16]. However, it is also valuable for subjects at the early stages of the disease, as the excessive and non-personalized cognitive load may negatively interfere with the motivational aspects, even when cognitive functions are not significantly affected.

A critical point is to verify the effectiveness of this new type of monitoring–rehabilitation protocol through dedicated studies involving parkinsonian subjects with different stages of the disease, cognitive impairment, motivations, and attitudes toward technological devices. Indeed, a longitudinal study involving subjects at different stages of the disease might allow, for instance, to evaluate through the PoS task, balance improvements due to periodical training, as these could not be observed from the protocol defined for this preliminary work. Moreover, a longitudinal study could allow us to also evaluate the effects of pharmacological treatments (e.g., correlation with last Levodopa intake), as one of the goals of this system may also be to support clinicians in tuning drug therapy according to the subject’s performance during evaluative tasks and exergames.

At present, the study has some weaknesses. The sample size is small, so further studies on a larger number of subjects will be needed to confirm and consolidate the results presented. Another drawback is that the system is usable only indoors, as the RGB-Depth camera, which works in the infrared spectrum, may be affected by outdoor sunlight. Approaches based only on RGB streams and innovative deep-learning libraries for body tracking could allow researchers to overcome this limitation, with the disadvantage, however, of working in two-dimensional space. Regarding robustness, several recent studies have proved the body tracking accuracy of the Azure Kinect, even in comparison with gold standard systems for movement analysis [71,72,73,74,75,76]. However, it is important to pay attention to clothing, particularly loose clothing, as it could interfere with the depth map estimation, and consequently with the accuracy of the skeletal model.

Nevertheless, the preliminary results presented in this study confirm the expected initial goals and encourage us to pursue this line of research. The forthcoming experimental tests in unsupervised settings will also include a deeper evaluation of usability and acceptability through questionnaires and interviews, to collect positive and negative feedback from the participants and further empower the proposed solution.

## 5. Conclusions

The paper proposes a vision-based solution for PD subjects that integrates evaluation motor tasks and virtual rehabilitative/training exergames, exploiting the potential of the novel Azure Kinect camera and its noninvasive body tracking algorithm. Preliminary results indicate that the system is capable of quantifying functional parameters related to evaluative motor tasks and virtual exergames, detecting statistically significant differences in motor performance between healthy and PD subjects. However, the most innovative feature of the proposed solution is the integration of evaluative and rehabilitative aspects. In this study, we have demonstrated the potential alternative use of exergames. For example, exergames could be used as a tool to propose evaluative motor tasks in a more fun and engaging environment (comparison between the traditional and gamified version of LA task), using the game configuration parameters to train those motor components that are not properly activated in the traditional assessment (e.g., poor range or speed of movement). Or the use of exergames to get immediate benefits on specific motor functions measured through evaluative motor tasks (as in the case of arm swing improvement during walking). Anyway, this is only the first step toward the implementation of a closed loop between evaluative motor tasks and exergames that would allow, on the one hand, to automatically adapt the difficulty of exergames to the current motor condition measured through evaluative motor tasks, and on the other hand, to automatically evaluate the effects of exergames on the motor condition and adapt the rehabilitation protocol consequently and appropriately. In addition, the main features of the proposed solution (in particular, noninvasiveness and usability) make the system suitable for home monitoring and rehabilitation; this could allow, in the near future, the definition of new follow-up protocols that use innovative technological approaches to support traditional clinical methodologies.

## Figures and Tables

**Figure 1 sensors-22-08173-f001:**
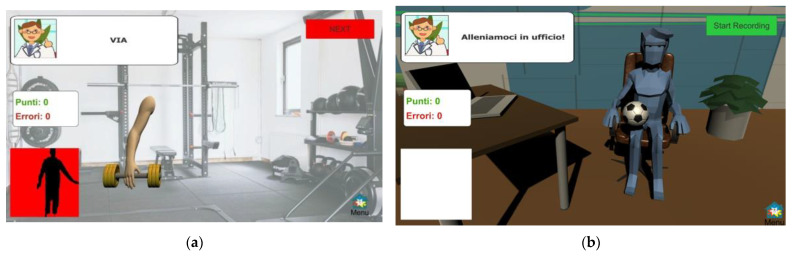
Screenshots of the main game scenes for the LWL/FWL (**a**) and BB (**b**) exergames during a patient’s game session. Textual messages and scoring board are used to guide the user’s execution.

**Figure 2 sensors-22-08173-f002:**
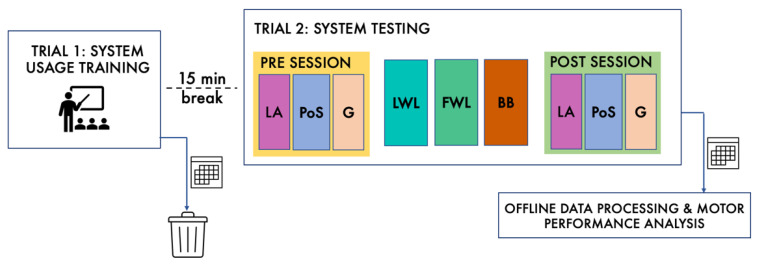
Schematic summary of the experimental protocol: during Trial 1, subjects were trained in using the system (collected data were discarded); Trial 2 consisted of two evaluative sessions before (PRE) and after (POST) testing of the exergames (collected data were offline processed and analyzed to evaluate motor performance at the end).

**Figure 3 sensors-22-08173-f003:**
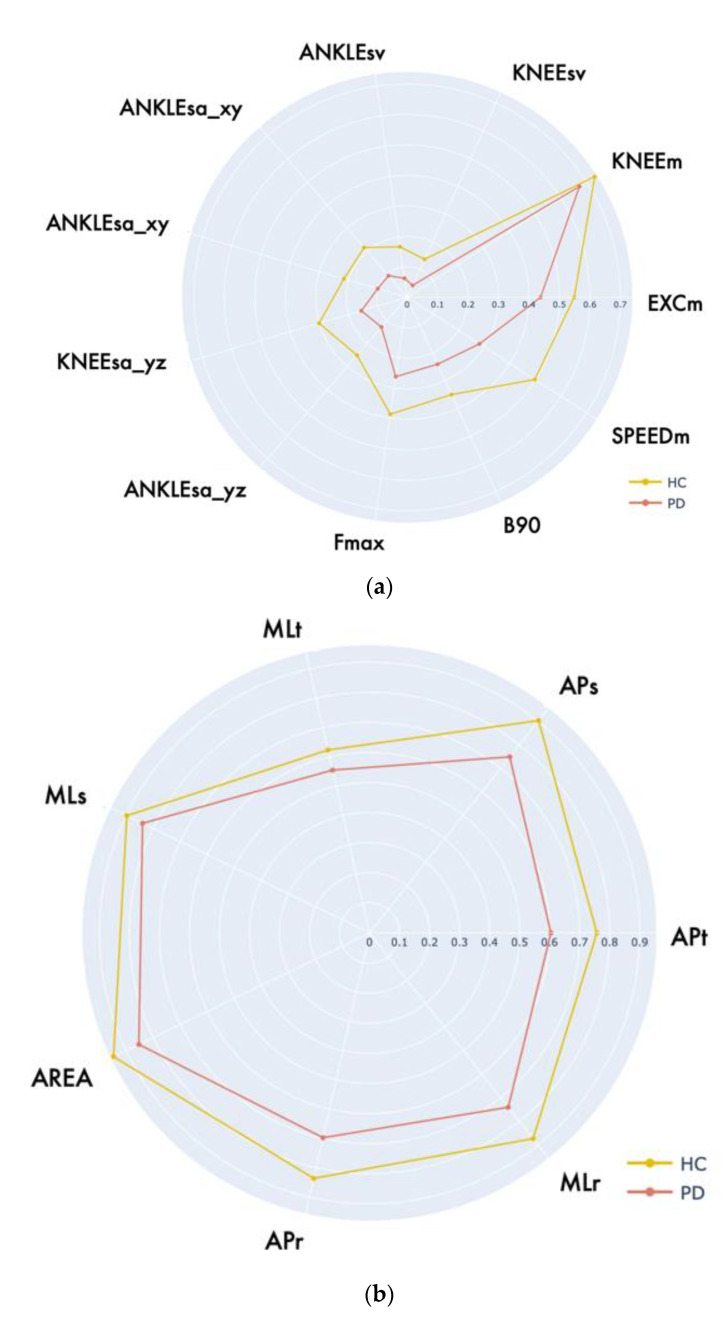
Radar charts of the normalized parameters of PD and HC subjects for all the UPDRS motor tasks: (**a**) leg Agility task; (**b**) postural Stability task; (**c**) gait task.

**Figure 4 sensors-22-08173-f004:**
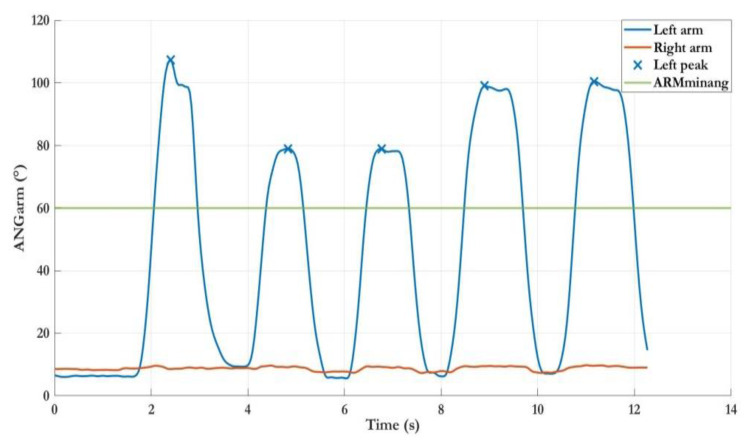
Example of the trajectories of left and right arm angles during the first subtask of the Lateral Weightlifting exergame (left arm movements). Correctly, no movement was detected in the right arm. All movements performed with the left arm were considered good movements (blue crosses).

**Figure 5 sensors-22-08173-f005:**
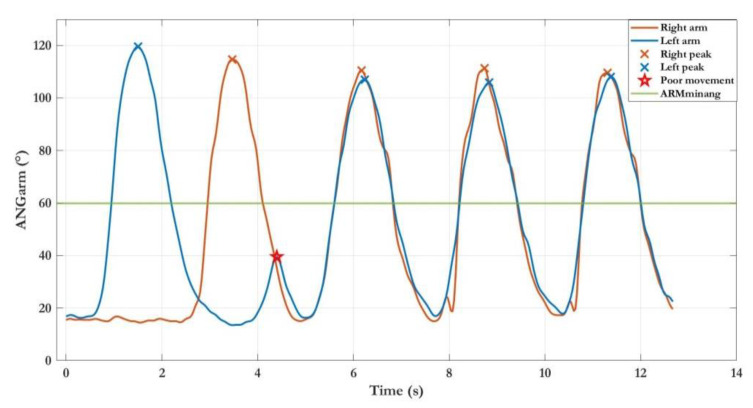
Example of the trajectories of left and right arm angles during the third subtask of the Frontal Weightlifting exergame (simultaneous movements). Anomalies detected: PM (red star) and non-simultaneous movements (first blue and orange crosses). The last three movements were correctly performed.

**Figure 6 sensors-22-08173-f006:**
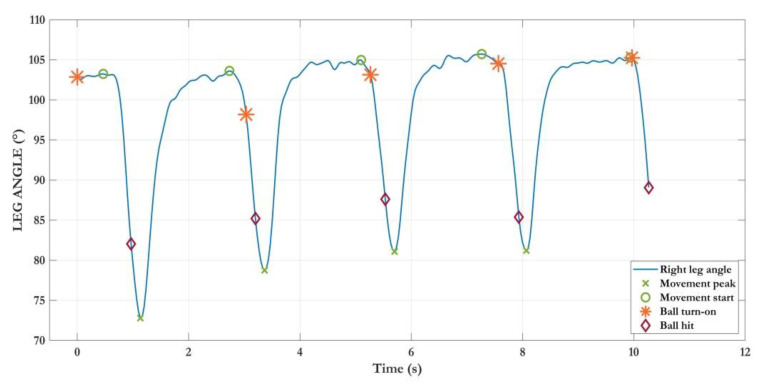
Example of the trajectory of right leg angle during the first subtask of Bouncing Ball exergame. The subject eventually learns to wait for ball light-ons before starting to move (green circles approaching orange stars in the last movements).

**Figure 7 sensors-22-08173-f007:**
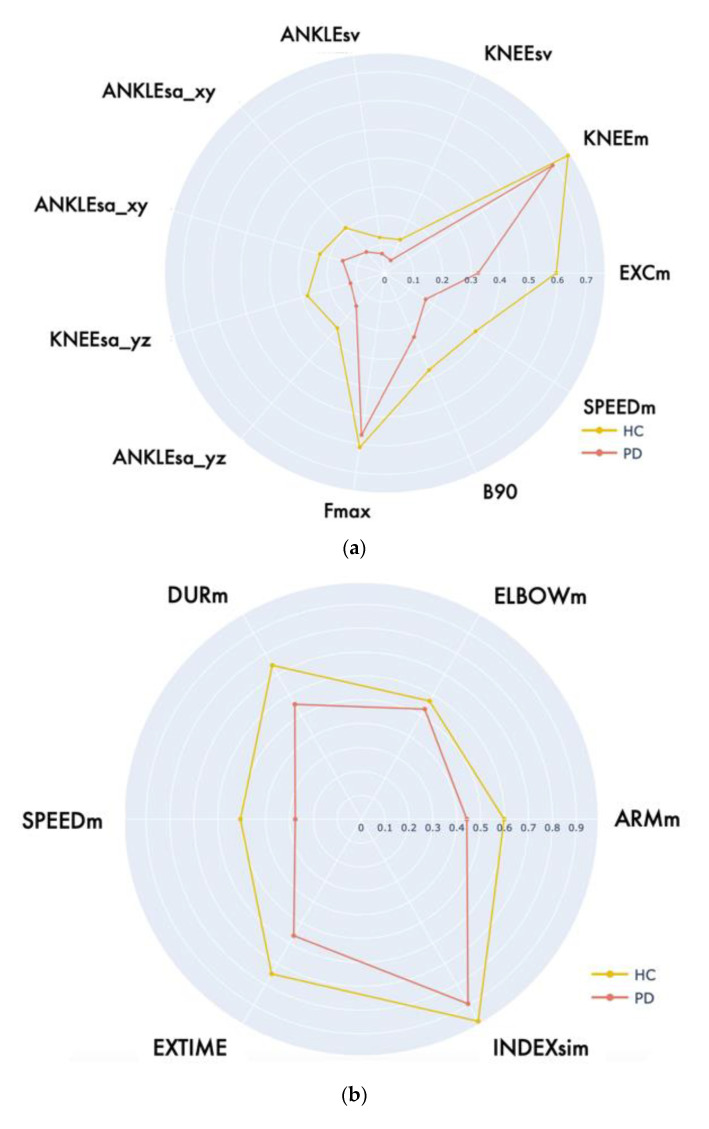
Radar charts of the normalized parameters of PD and HC subjects for all the exergames: (**a**) Bouncing Ball; (**b**) Lateral Weightlifting; (**c**) Frontal Weightlifting.

**Figure 8 sensors-22-08173-f008:**
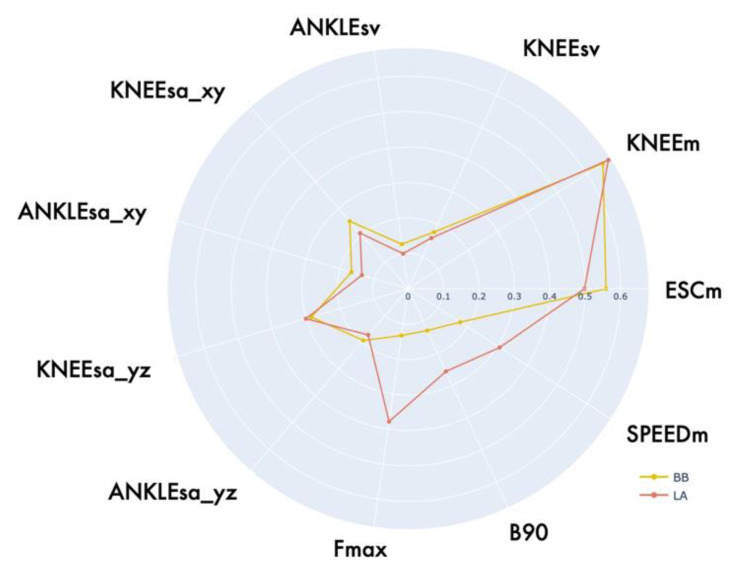
Radar chart comparison between functional parameters estimated during LA and BB executions for PD group.

**Figure 9 sensors-22-08173-f009:**
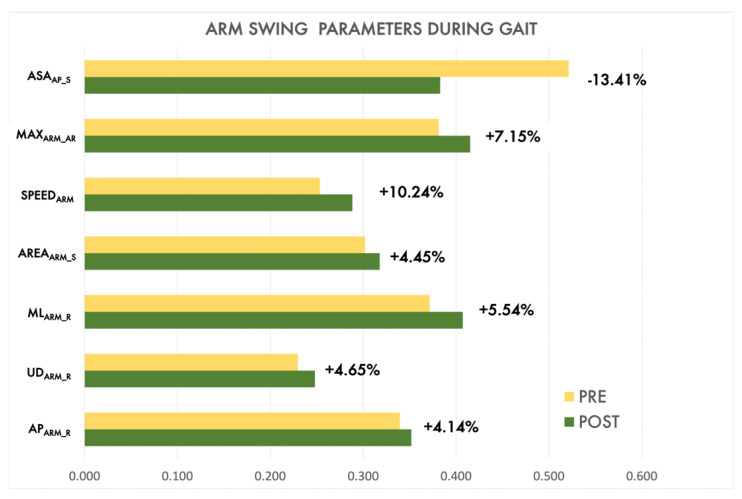
Normalized bar plots comparison of arm swing features (G task) between PRE and POST sessions by the PD group, after executing Lateral Weightlifting and Frontal Weightlifting.

**Table 1 sensors-22-08173-t001:** Estimated parameters for motor tasks and exergames.

Task	Parameter	Meaning	Unit
LA/BB	EXC_M_	Mean of ANG_LEG_ excursions	deg
	KNEE_M_	Mean ANG_KNEE_ during motion	deg
	ANKLE_SA_YZ_	ANKLE sway area (Frontal)	cm^2^
	KNEE_SA_YZ_	KNEE sway area (Frontal)	cm^2^
	ANKLE_SA_XY_	ANKLE sway area (Transverse)	cm^2^
	KNEE_SA_XY_	KNEE Sway Area (Transverse)	cm^2^
	ANKLE_SV_	ANKLE joint sway volume	cm^3^
	KNEE_SV_	KNEE joint sway volume	cm^3^
	SPEED_M_	Mean of leg movements speed	deg/s
	F_MAX_	Frequency at the maximum power	Hz
	B90	Frequency at 90% of the power	Hz
PoS	AP_R_	Range of antero-posterior (AP) sway	cm
	AP_T_	Total antero-posterior sway	cm
	AP_S_	Maximum antero-posterior sway speed	cm/s
	ML_R_	Range of medio-lateral (ML) sway	cm
	ML_T_	Total medio-lateral sway	cm
	ML_S_	Maximum medio-lateral sway speed	cm/s
	AREA	Sway area (AP-ML)	cm^2^
G	CADENCE	Number of steps per minute	step/min
	DSUP	Duration of double support	s
	STANCE	Stance duration (%gait cycle)	%
	STEP_L_	Mean of step length	m
	SPEED_WALK_	Mean gait speed	m/s
	SPEED_ARM_	Maximum speed on AP	cm/s
	AP_ARM_R_	Range of antero-posterior arm sway	cm
	ML_ARM_R_	Range of medio-lateral arm sway	cm
	UD_ARM_R_	Range of up-down arm sway	cm
	AREA_ARM_S_	Sway area of arm (AP-ML)	cm^2^
	MAX_ARM_AR_	Maximum arm angle range	deg
	ASA_AP_S_	Asymmetry of antero-posterior arm sway	%
LWL/FWL	ARM_M_	Mean of maximum angle peaks of ANG_ARM_	deg
	ELBOW_M_	Mean of ANG_ELBOW_	deg
	DUR_M_	Mean of arm movements duration	s
	SPEED_M_	Mean of arm movements speed	deg/s
	INDEX_SIM_	Index of simultaneity (only last level)	-
	EXTIME	Time to complete the exercise	s

**Table 2 sensors-22-08173-t002:** Number of valid executions included in data analysis (in parenthesis, the number of discarded executions).

Exercise	PD Executions	HC Executions
LA	75 (5)	59 (1)
PoS	40 (0)	30 (0)
Gait	40 (0)	30 (0)
LWL	53 (7)	42 (3)
FWL	55 (5)	42 (3)
BB	36 (4)	28 (2)

**Table 3 sensors-22-08173-t003:** Median and percentiles (first and third) related to each evaluative motor task for PD and HC groups, with test statistic, *p*-value, and significance level.

		Median (1° and 3° Percentiles)		Mann–Whitney	
Task	Parameter [Unit]	PD Group	HC Group	Statistic	*p*-Value
LA	EXC_M_ [deg]	28.24 (22.64, 34.30)	36.01 (22.23, 41.90)	643.00	0.062
	KNEE_M_ [deg]	99.34 (88.92, 109.78)	101.94 (99.40,106.63)	735.00	0.277
	ANKLE_SA_YZ_ [deg]	120.05 (88.80, 165.76)	168.52 (165.75, 279.11)	567.00	0.010 *
	KNEE_SA_YZ_ [cm^2^]	55.37 (38.17, 87.51)	86.51 (57.83, 121.38)	555.00	0.009 **
	ANKLE_SA_XY_ [cm^2^]	245.19 (130.52, 394.41)	451.42 (249.79, 854.70)	559.00	<0.001 ***
	KNEE_SA_XY_ [cm^2^]	60.13 (50.29, 79.23)	113.40 (76.81, 147.22)	412.00	<0.001 ***
	ANKLE_SV_ [cm^3^]	245.19 (130.52, 394.41)	451.42 (249.78, 854.70)	521.00	0.001 **
	KNEE_SV_ [cm^3^]	117.38 (73.38, 211.05)	239.38 (131.69, 521.61)	481.00	0.001 **
	SPEED_M_ [deg/s]	71.42 (46.41, 94.10)	108.73 (86.18, 129.37)	367.00	<0.001 ***
	F_MAX_ [Hz]	0.98 (0.62, 1.40)	1.32 (0.93, 2.19)	572.00	0.014 *
	B90 [Hz]	1.34 (0.98, 1.76)	1.71 (1.16, 2.67)	619.50	0.039 *
PoS	AP_R_ [cm]	2.47 (1.74, 3.93)	1.75 (1.27, 2.16)	269.00	0.001 **
	AP_T_ [cm]	29.70 (24.60, 33.1)	23.40 (20.90, 26.70)	256.00	<0.001 ***
	AP_S_ [cm/s]	5.40 (4.59, 6.68)	4.05 (3.44, 4.44)	190.00	<0.001 ***
	ML_R_ [cm]	1.55 (0.98, 2.14)	0.87 (0.50, 1.46)	277.00	0.002 **
	ML_T_ [cm]	16.00 (13.90, 19.90)	15.40 (12.00, 18.70)	418.00	0.232
	ML_S_ [cm/s]	3.75 (2.97, 4.54)	3.26 (2.67, 3.65)	368.00	0.061
	AREA [cm^2^]	1.97 (1.05, 3.93)	0.85 (0.52, 1.99)	265.00	<0.001 ***
G	CADENCE [step/min]	100.67 (87.76, 117.31)	108.11 (94.94, 115.38)	972.00	0.144
	DSUP [s]	0.34 (0.24, 0.50)	0.17 (0.11, 0.31)	612.00	<0.001 ***
	STANCE [%]	63.88 (61.84, 70.98)	59.28 (56.44, 62.89)	564.00	<0.001 ***
	STEP_L_ [m]	0.55 (0.50, 0.58)	0.66 (0.61, 0.71)	324.00	<0.001 ***
	SPEED [m/s]	0.91 (0.71, 1.02)	1.17 (1.01, 1.27)	480.00	<0.001 ***
	SPEED_ARM_ [cm/s]	32.72 (21.53, 57.19)	60.60 (29.67, 79.98)	757.00	0.004 **
	AP_ARM_R_ [cm]	15.11 (9.71, 21.74)	20.73 (12.02, 28.17)	813.00	0.012 *
	ML_ARM_R_ [cm]	5.11 (4.00, 6.57)	4.76 (4.11, 5.92)	1080.00	0.566
	UD_ARM_R_ [cm]	4.66 (3.14, 6.06)	5.41 (4.32, 7.10)	816.00	0.013 *
	AREA_ARM_S_ [cm^2^]	40.57 (23.18, 76.53)	61.01 (40.71, 86.52)	865.00	0.033 *
	MAX_ARM_AR_ [deg]	14.96 (6.18, 21.94)	21.82 (16.90, 26,88)	756.00	0.004 **
	ASA_AP_S_ [%]	−14.98 (−18.3, −7.46)	−6.55 (−11.74, −2.72)	704.00	<0.001 ***

***: *p*-value < 0.001; **: *p*-value < 0.01; *: *p*-value < 0.05.

**Table 4 sensors-22-08173-t004:** Median and percentiles (first and third) estimated for PD and HC groups, for FWL, LWL, and BB exergames, with test statistic, *p*-value, and significance level.

		Median (1° and 3° Percentiles)		Mann–Whitney	
EXERGAME	Parameter [Unit]	PD Group	HC Group	Statistic	*p*-Value
BB	EXC_M_ [deg]	29.00 (24.01, 33.97)	45.20 (29.76, 48.26)	135.00	<0.001 ***
	KNEE_M_ [deg]	98.68 (88.87, 108.45)	99.99 (97.17,104.83)	283.00	0.671
	ANKLE_SA_YZ_ [cm^2^]	158.31 (59.74, 234.09)	218.86 (132.33, 331.93)	192.00	0.030 *
	KNEE_SA_YZ_ [cm^2^]	54.29 (32.91, 89.11)	93.77 (76.10, 160.04)	149.00	0.002 **
	ANKLE_SA_XY_ [cm^2^]	59.32 (38.53, 88.61)	98.21 (65.69, 124.67)	170.00	0.009 **
	KNEE_SA_XY_ [cm^2^]	66.34 (50.07, 97.58)	124.52 (77.09, 156.68)	154.00	0.003 **
	ANKLE_SV_ [cm^3^]	323.26 (106.64, 516.91)	553.44 (388.48, 775.76)	189.00	0.025 **
	KNEE_SV_ [cm^3^]	117.93 (68.18, 259.63)	284.29 (139.39, 498.37)	170.00	0.009 **
	SPEED_M_ [deg/s]	38.69 (34.31, 52.28)	81.79 (52.83, 92.77)	83.00	<0.001 ***
	F_MAX_ [Hz]	0.44 (0.34, 0.45)	0.44 (0.39, 0.49)	256.00	0.335
	B90 [Hz]	0.83 (0.67, 1.03)	1.123 (0.98, 1.37)	158.00	0.005 *
LWL	ARM_M_ [deg]	97.00 (89.60, 105.00)	108.00 (104.00, 114.00)	764.00	0.001 **
	ELBOW_M_ [deg]	147.00 (141.00, 150.00)	149.00 (144.00, 153.00)	1519.00	0.244
	DUR_M_ [s]	2.32 (2.04, 2.82)	1.95 (1.58, 2.13)	708.00	<0.001 ***
	SPEED_M_ [deg/s]	74.00 (63.60, 87.70)	106.00 (85.60, 148.00)	567.00	<0.001 ***
	INDEX_SIM_ [-]	0.030 (0.021, 0.042)	0.022 (0.017, 0.023)	39.5	0.004 **
	EXTIME [s]	12.2 (10.5, 15.2)	10.5 (8.75, 11.6)	839.00	<0.001 ***
FWL	ARM_M_ [deg]	110.07 (98.25, 119.92)	120.28 (102.63, 142.74)	1141.00	0.004 **
	ELBOW_M_ [deg]	144.75 (138.77, 149.53)	145.58 (137.87, 149.59)	1675.00	0.981
	DUR_M_ [s]	2.50 (2.17, 2.91)	1.81 (1.50, 2.11)	494	<0.001 ***
	SPEED_M_ [deg/s]	79.48 (62.53, 101.13)	139.47 (91.86, 183.13)	627.00	<0.001 ***
	INDEX_SIM_ [-]	0.030 (0.020, 0.042)	0.021 (0.016, 0.023)	52.50	0.038 **
	EXTIME [s]	14.06 (11.83, 16.35)	10.3 (10.37, 11.86)	611.00	<0.001 ***

***: *p*-value < 0.001; **: *p*-value < 0.01; *: *p*-value < 0.05.

## Data Availability

Data are available on request.

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
