# Peer review of "Assessment Tasks and Virtual Exergames for Remote Monitoring of Parkinson’s Disease: An Integrated Approach Based on Azure Kinect"

_sensors, 2022, doi:10.3390/s22218173_

Round 1

Reviewer 1 Report

Dear Authors,  

Apart from the fact that the number of participants in the study is small , I have no objections to the study and I suggest to accept it in a present form.

Reviewer 2 Report

Dear authors, 

Congratulations on the article submitted to Sensors journal. 

The quality and presentation is excellent. 

The results found and the design of the study are of great use to clinicians and rehabilitation teams. 

I only suggest a small contribution: 

Include a flow chart in 2.5 to visualize the design and phases of the study. 

Kind regards, 

Reviewer 3 Report

Dear authors,

After the review process, I have several comments: the abstract should contain numerical data; in the last part of the introduction, you should eliminate results; the authors should present how they realized figure 1 and add o clear explanation in the legend; in figure 8 should add S.D. and delete the numbers; in the last part of the discussion section, you should add additional data related with the therapeutic applications of biomolecules along with a target in silico approach in neurodegenerative diseases.

Best regards!

Reviewer 4 Report

Comments to Author (General)

·       The authors shall have to check the arrangement of section 2. (For example, 2.1, 2.2, 2.3, and so on). Section 2.2 is missing.

·       The authors shall have to rewrite the abstract based on the sequential discussions on problem statement, research methods, specific outcomes and results confirmation with some verifications.

·       The authors shall have to give the appropriate citation in the manuscript in details.

·       The authors shall have to modify the high resolution pictures of Figure.2, 6, and 7.

Comments to Authors (Specific)

·       In abstract, the authors mentioned “closed loop to design” and the details discussions on that approaches shall have to be mentioned in recommendation section after discussions.

·       According to the presentation in “Introduction”, the authors shall have to draw the overall system block diagram for proposed design.

·        HCI and GUI are very effective and efficient techniques, the authors shall have to discuss on such kind of interfacing system with specific language in details in respective section.

·       The authors shall have to give the mathematical expressions or modellings for numerical analyses with MATLAB procedures and it is very useful for analysis.

·       The authors shall have to discuss how to specify the normality hypothesis (p<0.05) in Table 1 for derivative for analysis.

·       The authors shall have to give the reason why 95% is significant level for the consideration of the test.

·       The authors shall have to give the discussion on “poor movement” in Figure.4.

·       The authors shall have to discuss in details on Figure.7.

·       The authors shall have to interpret the first bar plots of ASA in Figure.8.

·       The authors shall have to confirm and verify that the statistic table for current analysis and recent works before conclusion.

·       The limitations and robustness of the developed system shall be expressed in details.

Round 2

Reviewer 3 Report

no other comments